# Drug Delivery Systems of Natural Products in Oncology

**DOI:** 10.3390/molecules25194560

**Published:** 2020-10-06

**Authors:** Marisa Colone, Annarica Calcabrini, Annarita Stringaro

**Affiliations:** National Center for Drug Research and Evaluation, Istituto Superiore di Sanità, Viale Regina Elena 299, 00161 Rome, Italy; marisa.colone@iss.it (M.C.); annarica.calcabrini@iss.it (A.C.)

**Keywords:** multidrug resistance, chemotherapy, nanomedicine, drug delivery systems, natural products

## Abstract

In recent decades, increasing interest in the use of natural products in anticancer therapy field has been observed, mainly due to unsolved drug-resistance problems. The antitumoral effect of natural compounds involving different signaling pathways and cellular mechanisms has been largely demonstrated in in vitro and in vivo studies. The encapsulation of natural products into different delivery systems may lead to a significant enhancement of their anticancer efficacy by increasing in vivo stability and bioavailability, reducing side adverse effects and improving target-specific activity. This review will focus on research studies related to nanostructured systems containing natural compounds for new drug delivery tools in anticancer therapies.

## 1. Introduction: Limitations of Anticancer Therapies

Cancer is among the leading causes of death worldwide. The number of new cancer cases per year is expected to rise to 23.6 million by 2030 [1]. In Europe there were an estimated 3.91 million new cases of cancer (excluding non-melanoma skin cancer) and 1.93 million deaths from cancer in 2018. The most common cancer sites were cancers of the female breast (523,000 cases), followed by colorectal (500,000), lung (470,000) and prostate cancer (450,000). These four cancers represent half of the overall burden of cancer in Europe. The most common causes of death were lung (388,000 deaths), colorectal (243,000), breast (138,000) and pancreatic (128,000) cancers [2]. The late diagnosis and non-responsive therapy represent the main causes of higher mortality among many cancer patients.

In recent years a better understanding of the mechanisms underlying tumor biology led to a significant progress in the prevention, detection and treatment of cancer. Traditional approaches for cancer treatment include surgery, radiation therapy, chemotherapy, immunotherapy and targeted and hormone therapy [3]. Unfortunately, sometimes these approaches are limited because they show low specificity, as they can also affect healthy cells and/or the immune system, then causing negative side effects. In addition, all therapies used in cancer treatment, with the exception of surgery, can induce a drug-resistance response in tumor cells (Figure 1). For these reasons, anticancer therapy research is continuously aimed at searching for more effective therapeutic approaches [4].

Drug resistance is still a great problem in cancer therapy. Resistance can be intrinsic or acquired when there is low or no response to anticancer therapy from the beginning or during the course of therapy, respectively. One-drug resistance refers to resistance to one specific drug; when patients show resistance to one drug and become resistant to other unrelated drugs (with different structures and mechanisms of action), the term multidrug resistance (MDR) is employed [5].

A number of factors can contribute to clinical MDR in cancer: factors related to the host as well as factors strictly associated to tumor mass and its interaction with surrounding tissues (Figure 1) [6].

Among host factors, genetic variants, interactions between different drugs and alteration in the pharmacokinetics are the most important. The host genetic variants are mainly related to genes encoding drug-metabolizing enzymes, drug uptake/efflux transporters, drug targets, DNA repair and cell cycle control mechanisms [7]. When different drugs are administered to a cancer patient (such as anti-emetics, analgesics, anxiolytics), interactions between them can induce a synergistic, antagonist or additive response [8]. As a direct consequence of these interactions, alteration in drug activity can occur, contributing to reduce its therapeutic efficacy. In addition, pharmacokinetic processes, related to the drug absorption, distribution, metabolism and excretion mechanisms, represent additional host-related factors that strongly affect drug efficacy, as they can be responsible for reduced the drug amount and/or drug activity in the target site [6].

Among tumor factors, alterations in the intracellular drug concentration is one of the most important. The amount of a drug accumulated in drug-resistant cells is usually reduced with respect to sensitive cells. This event can be due to a reduced drug diffusion/permeability or to the activity of transmembrane proteins belonging to the ATP-binding cassette (ABC) transporter superfamily. These proteins are often overexpressed and are able to extrude structurally different compounds across plasma and intracellular membranes, through an active mechanism coupled to ATP hydrolysis [9]. In different types of tumors, the induction of drug resistance has been directly correlated to the overexpression of ABC proteins as P-glycoprotein (P-gp), multidrug-resistance-associated protein 1 (MRP1) and breast cancer resistance protein (BCRP). The overexpression of P-gp, one of the best studied drug transporters associated with MDR, is responsible for the resistance phenotype of a number of neutral and cationic hydrophobic antitumor drugs (paclitaxel, docetaxel, etoposide, vinblastine, vincristine, doxorubicin, daunorubicin, actinomycin D, gefitinib, sunitinib) [10,11,12]. Consequently, in order to overcome this form of MDR, in recent years, a lot of studies have been aimed at finding substances, including natural ones, able to inhibit ABC transporter activity increasing intracellular drug concentration [13]. Unfortunately, molecules able to efficiently inhibit in vitro ABC proteins, when tested in clinical trials, often induce high toxicity and do not overcome in vivo MDR phenomena [14]. To avoid these limitations, therapeutic approaches are increasingly based on the use of monoclonal antibodies against ABC transporters as well as delivery systems containing molecules with MDR modulator activity or siRNA/microRNA able to downregulate drug transporter transcription [15].

Other tumor factors that can exert a pivotal role in inducing clinical MDR are: (1) deregulation of cell death mechanisms (apoptosis, autophagy and anoikis); (2) alteration of DNA damage response and repair mechanisms; (3) epigenetic modifications (such as DNA methylation, histone and chromatin alterations, deregulation of microRNAs); (4) tumor microenvironment (TME) [7].

TME is a dynamic network of cancer cells, extracellular matrix components and stromal cells (cancer-associated fibroblasts, endothelial cells, immune system cells). TME is characterized by a reduced supply of nutrients and oxygen and low pH. Hypoxic conditions contribute to the reduced development of vessels within the tumor mass, thus causing a decreased drug amount in target sites and, consequently, drug efficacy. All these elements strongly suggest that TME plays a crucial role in cell response to chemo, radio and targeted therapies [16]. Another microenvironment factor that can contribute to tumor complexity and heterogeneity, and indirectly to therapy response, is represented by the presence of a population of tumor cells with stem features.

Cancer stem cells (CSCs), initially identified in AML and later in solid tumors (such as breast, prostate, brain, colon, melanoma), represent a very small subpopulation in a tumor and possess the ability of self-renewal and to form new tumors in nude mice [17]. CSC biological features are very similar with those of MDR cells; they usually show intrinsic heterogeneity, plasticity, overexpression of drug ATP transporters, more efficient DNA repair mechanisms, high levels of metabolic regulators (ROS scavengers, ALDH) and slow or no proliferation (quiescence, dormancy, senescence) [18]. These features contribute to CSC survival and, consequently, to their ability to develop secondary tumors, strongly supporting the hypothesis that often clinical therapeutic failure can result from inefficient CSCs targeting inside the tumor mass [19].

In addition to stemness and tumor heterogeneity, epithelial-to-mesenchymal transition (EMT) has been associated with resistance to anticancer therapies, although a direct correlation has been demonstrated mostly in cellular and preclinical models and, to a lesser extent, in clinical samples. During tumor progression, EMT is responsible for the coexistence of different cellular phenotypes showing both epithelial and mesenchymal features: cancer cells tend to lose intercellular contacts, as well as apical-basal polarity, detach from extracellular matrix components and acquire mesenchymal features (expression of N-cadherin, vimentin, fibronectin, matrix metalloproteinases). Thus, these hybrid cells can also show invasive and stem cell-like features, be adaptable to microenvironment changes and contribute to the onset of tumor resistance to anticancer therapy and immune escape [20].

Despite the positive outcomes in the field of cancer research, considering all the above factors responsible for limited chemotherapy success, a lot of studies are continuously concerned to develop more efficient therapeutic strategies, in order to selectively target cancer cells, avoid a MDR response, overcome biological barriers and achieve a spatial, temporal and dose control of drug release.

Natural products, either in their naturally occurring forms or their synthetically modified forms, have been demonstrated over 40 years to be an important source for cancer preventive and chemotherapeutic agents. Indeed, considering the period between the 1940s and the end of 2014, almost 50% of all small molecules approved for cancer therapy is represented by natural products or compounds that are directly developed from them [21]. Unfortunately, the use of most natural products in anticancer therapy, as well as against infections or other diseases, is limited due to their low bioavailability, directly related to both their lipophilic and hydrophilic nature [22], and to the possible induction of cytotoxic effects [23]. Nanomedicine-based strategies allow to improve the bioavailability of many natural compounds as well as to increase their selective activity against cancer cells.

## 2. Nanomedicine and Drug Delivery

Nanomedicine (Figure 2) has received increasing attention for its ability to improve efficacy of cancer therapeutics, reduce side effects and overcome drug resistance-related problems [24].

Nanoparticles (NPs) or nanocarries (NCs) used in medicines are increasingly considered as potential candidates to safely carry therapeutic agents into targeted compartments in an organ, particular tissue or cell. NPs are a broad class of materials intended for a broad spectrum of clinical applications (medical devices, components of vaccine formulations or drug carriers for therapeutic intervention of inflammatory, viral and cancer diseases) [25]. A number of nano-microdelivery systems are designed to encapsulate the drug in carriers (liposomes, polymeric nanocapsules and dendrimers), which mask the unfavorable biopharmaceutical properties of the molecule and replace them with the properties of materials used for the nano-delivery system. Advances in nanomedicines are also applied for site-specific delivery of drugs, including natural products. In this context, it is very important to include the research on nanoparticulate delivery systems in the preformulation development of new drug products. To pursue this approach, biomaterial science has stepped into the formulation of smart materials and miniaturized drug delivery devices.

The base knowledge of NPs and the assessment of their safety and efficacy are still expanding. A wide variety of classes are being investigated as nanocarriers for cancer treatment, including lipid-based, polymer-based, inorganic nanoparticles and drug-conjugated nanoparticles (Figure 3).

Some of these tools have also been approved for use in clinical protocols [26]. In this section, the main categories of NPs are presented.

### 2.1. Organic Nanoparticles

#### 2.1.1. Lipid-based NPs

Liposomes and lipid nanocapsules (LNCs) are two nanodelivery systems based on lipids. Liposomes, constituted by a hydrophilic “core” surrounded by lipid bilayers, are able to incorporate hydrophobic drugs within the lipid bilayer or hydrophilic drugs within the aqueous core [27]. Since the structure of the lipid bilayer is similar to the cell membrane, the liposomal nanoparticle can either fuse with the cell membrane or lyse once combined with intracellular organelles, thus releasing the drug. Furthermore, the ability of liposomes to co-encapsulate both therapeutic and diagnostic agents opens the way for a novel application of liposomal delivery systems as theranostic platforms [28,29,30]. The delivery of liposomes to cancer cells often relies on passive targeting and is based on the enhanced permeability and retention effect [31]. Moreover, the addition of polyethylene glycol (PEG, known as stealth liposomes) is able to increase their circulation time. A number of liposome-based systems with different targeting ligands are already undergoing clinical trials, such as an anti-HER2 mAb with Paclitaxel [32] and a mAb 2C5 with Doxorubicin (Doxil^®^) [33]. A triggered release of the drug internalized into liposomes has been achieved by an external stimulus as hyperthermia with ThermoDox^®^, recently approved by U.S. Food and Drug Administration [34,35].

Another type of lipid-mediated delivery system is represented by lipid nanocapsules (LNCs) that are surrounded by a membrane composed of PEGylated surfactants [36]. The encapsulation within LNCs of some antitumoral drugs, such as doxorubicin and paclitaxel, have shown good results both in vitro and in vivo, as demonstrated by the higher intracellular drug delivery and reduced tumor size observed when LNC formulations were administered.

#### 2.1.2. Exosomes

In multicellular organisms, cells communicate via extracellular molecules such as nucleotides, lipids, short peptides and proteins. Released by cells, these molecules can bind to receptors of other cells, inducing intracellular signaling and physiological modification of the recipient cells. In addition to these single molecules, eukaryotic cells also release biological micro- and nano-structures in the extracellular environment, called membrane vesicles, containing lipids, proteins and even nucleic acids, which affect the encounter cells in a complex way [37]. Exosomes are equivalent to cytoplasm enclosed in a lipid bilayer with the transmembrane proteins localized in the cellular surface. They are formed inside the cells in compartments known as multi-vesicular bodies (MVBs), which take up bits of the cytoplasm and its contents into membrane-bound vesicles. Once the MVBs are fused with the plasma membrane, these internal vesicles are secreted [38]. The biological function of exosomes is still under study; they can mediate intercellular communication or induce intra- and extracellular signals, and it is well known that exosomes are involved in the exchange of functional genetic information [39,40].

Exosomes are secreted by most cell types (B-cells, dendritic cells, T-cells, mast cells, epithelial cells, platelets, stem cells and cancer cells) in normal and pathological conditions. They have been found in cell culture media, in different biological [41,42,43,44] and physiological fluids, such as serum, urine, breast milk, cerebrospinal fluid, bronchoalveolar lavage fluid, saliva and malignant effusions [45,46]. Due to the strong impact of exosomes in cancer pathogenesis and biological compatibilities (i.e., they are able to cross physiological barriers such as the Blood Brain Barrier), exosomes are strong candidates for advanced therapeutic applications. These biological features include targeting exosomes, re-engineering and modifying them as therapeutic devices. Drug loading of exosomes can be achieved either endogenously or exogenously. Endogenous, or passive loading is carried out by inducing the overexpression of the RNA molecules of interest in producer cells. This passive loading is accomplished by the native exosomal loading mechanisms of the cell itself and results in exosomes that contain the drug before their isolation. Exogenous or active loading starts with the collection of exosomes and requires co-incubation or electroporation of the exosomes with the drug/molecule of interest [47], such as siRNA [48], doxorubicin [49,50], paclitaxel [51] and curcumin [52], by using different strategies.

#### 2.1.3. Carbon-based Nanoparticles (Carbon Nanotubes and Graphene Nanoplatelets)

The fundamental characteristics of a system for an efficient administration of drugs are the control and precision of the process. The fast release of drugs would result in their incomplete absorption, gastrointestinal disorders and other side effects. It is also necessary that the drugs do not decompose during administration, as some components may be toxic on their own. Therefore, the drugs should be encapsulated within carriers made of compatible materials, in which they maintain their biological and chemical properties. Furthermore, carriers should decompose by dissolving in the solution or by excretory routes at the end of transport.

Carbon nanomaterials are mainly composed of carbon, usually in the form of hollow spheres, ellipsoids or tubes. Studies aimed at carbon nanomaterials have led to the discovery and development of carbon nanotubes (CNTs), which are made of graphite sheets (carbon atoms arranged in parallel planes) rolled up to form a cylindrical structure. The diameter of a nanotube is between 0.7 and 30 nm (larger objects are called carbon nanofibers). CNTs are very interesting due to their mechanical strength and electrical properties [53]. In addition, the ability to fill nanotubes with drugs, and to functionalize the surface with antigenic peptides, allows the development of innovative controlled transport systems for drug delivery-based therapies. To this aim, design strategies tend to obtain smaller and thicker objects to avoid inflammogenic effects and a better biodegradability. Indeed, they can be used for the administration of drugs, since they can reach cell nuclei without being recognized by cells thanks to their dimensions. CNTs can be immobilized with biocompatible materials and functionalized in order to be soluble in organic solvents and aqueous solutions [54]. Recently, different research groups have incorporated in nanotubes drugs such as doxorubicin and paclitaxel and nucleic acids, including antisense oligonucleotides and siRNAs [55]. In addition, several studies are investigating the possible use of nanotubes as contrast agents for imaging. Until now, no clinical trials have begun using CNT for the treatment or diagnosis of cancer, mainly due to their toxicity and similarity to asbestos fibers [56,57].

Recent discoveries on graphene, a two-dimensional, crystalline allotrope of carbon, stimulated research on related structures, such as pristine Graphite NanoPlatelets (GNPs), a 1–15 nm thick flake, constituted of 3–48 layers of graphene, obtained from Intercalated Graphite Compounds (GIC) via thermochemical exfoliation [58]. These novel nanomaterials opened new opportunities for biotechnological development thanks to their versatility. Due to the expanding applications of nanotechnology, human and environmental exposures to graphene-based nanomaterials are likely to increase in the future. However, the prospective use of graphene-based nanostructures in a biological context requires a detailed comprehension of their toxicity.

The use of graphene in therapeutics delivery is based on its high surface-area to volume ratio; polyaromatic structure can be functionalized, improving targeting to tissues. Moreover, the possibility to combine hydrophobic and hydrophilic regions on the surface of GNPs supports their interaction with lipids in cell membranes [59]. Recently, an alternative graphene form has been preferred to pristine graphene for biomedical applications. Graphene oxide (GO) is characterized by higher solubility thanks to its surface chemistry [60,61,62].

One of the earliest reports of graphene-based cancer drug delivery was by Yang et al. They obtained a graphene oxide-doxorubicin hydrochloride nanohybrid (GO-DXR) through a simple noncovalent method. Results showed an efficient high loading and pH-dependent release of the drug on GO, opening these novel GO-based nanohybrids for applications as drug carriers and biosensors [63]. Jokar et al. reported albumin-conjugated GO loaded with paclitaxel. The albumin-GO-paclitaxel complex showed pH-dependent release of the drug [64]. In another study, covalently conjugated PEG-GO structures were employed to deliver paclitaxel to A549 and MCF-7 cancer cell lines [65]. A faster drug accumulation inside the cells and a higher cytotoxic effect was observed with GO-PEG-PTX delivery system compared to free drug, thanking to the improved water solubility and bioavailability of the antitumoral drug.

#### 2.1.4. Polymeric Nanoparticles

Polymeric nanoparticles (NPs) are systems composed by natural, synthetic or semisynthetic polymers. The main features related to polymeric nanoparticles employment are the limited shape and wide size distribution. Polymeric NP are typically spherical, although their final size can be modulated during the synthesis process. Moreover, polymeric NP properties, as well as the amount, rate and pathway used for cellular uptake of the encapsulated drug molecule, may be modified in order to improve the final product [66]. As promising biomaterials, these NPs may overcome biological barriers, protect the therapeutic cargo and effectively deliver it to the diseased tissue. The problems associated with polymeric nanoparticles employment are related with organic solvents used for their formation, which could be present in the form of residues in the final formulation and provoke toxic effects.

Polymeric nanoparticles of natural origins are mainly made by polysaccharides, such as cellulose and lisozima [67]. Natural polymers can be chemically modified creating semisynthetic polymers, for example methylcellulose. Natural polymers extensively used in nanoparticle synthesis include chitosan, dextran, albumin, heparin, gelatin and collagen [68,69]. Chitosan-coated PLGA (poly(lactic-co-glycolic acid) NPs [70] and chitosan NPs [71,72,73] can carry and deliver proteins in an active form to specific organs.

Polymers utilized for nanoparticles of synthetic origin production are totally artificial, such as PLGA, PLA (Poly-lactic acid) and PMA (thiolated poly (methacrylic acid) [74,75]. Among these polymeric NPs, PLA-based nanoparticles showed a good potential in anticancer therapy. In a recent study, these pH and temperature-sensitive NPs induced no or low toxicity towards cell cultures and high stability at physiological conditions. When used as a nanovector for doxorubicin release, their three-dimensional (3D) supramolecular polymer network allowed to overcome drug resistance [76]. Among the polymer-based delivery systems, the albumin-based nanoparticle, protein-bound paclitaxel (Abraxane) is the only approved by the FDA for clinical use in the treatment of breast, non-small cell lung and pancreatic cancer [77,78]. Different albumin-based nanoparticles, such as ABI-008 [79], ABI-009 [80] and ABI-011 [81], are currently undergoing clinical trials. BIND-014 is the first PEG-PLGA targeted polymeric nanoparticle to reach phase I/II studies for the treatment of metastatic cancer and squamous cell non-small cell lung cancer. Its pharmaceutical activity is 10-fold higher than that of conventional docetaxel in tumor sites, and it prolongs the time the drug is maintained in the circulation [82].

### 2.2. Metallic and Magnetic Nanoparticles

Various forms of inorganic nanoparticles, including superparamagnetic iron oxides, gold nanoparticles and other metallic and non-metallic nanoparticles or nanoclusters, enhance the efficiency of radiotherapy and improve tumor imaging. Several of these inorganic nanoparticles (superparamagnetic iron oxides, gold nanoparticles) are sufficiently small (10–100 nm) to penetrate in distinct tissues through capillaries [83,84]. Others are larger and need to be delivered at disease-specific anatomic sites for passive targeting. Multifunctional inorganic NPs are also emerging as tools to target cancer [85,86,87]. Gold nanoparticles can be used to deliver small molecules such as proteins, DNA or RNA [88] Drugs can easily be attached through ionic or covalent bonds, or through adhesion. Additionally, PEG can be attached to the surface of metallic nanoparticles in order to increase stability and circulation time.

Such devices can contain the drug and also specific receptor-targeting agents, as antibodies or ligands, as well as magnetic resonance imaging contrast agents [89]. In early-stage clinical trials, some inorganic nanomaterials, such as gold nanoparticles and silica nanoparticles [90], showed negative effects, including toxicity and a lack of stability. NanoTherm, used for the treatment of glioblastoma, is the only one approved for clinical use. Tumors can be thermally ablated with NanoTherm by magnetic hyperthermia induced by entrapped superparamagnetic iron oxides nanoparticles. Additionally, Superparamagnetic iron oxide (Fe_3_O_4_) nanoparticles are under development as possible new strategies to deliver conjugated drugs using a local hyperthermia or oscillation strategies to targeted area. Moreover, magnetic fields can also be used to guide the drug within the body [91].

### 2.3. Active and Passive Target

The application of nanotechnology to drug delivery has an enormous potential concerning the improvement of selectivity in targeting neoplastic cells. Thanks to the enhanced permeability and retention (EPR) effect in tumor and inflamed tissue vasculature, chemotherapeutic drugs can be selectively delivered in specific regions [92]. Scarcely aligned endothelial cells in tumor vasculature result in preferred accumulation of nanocarriers in these tissues instead of healthy ones, improving the efficacy of the anticancer drug [86,93].

The EPR effect is based on two factors: the first exploits the fact that the endothelium around the blood vessels in the tumor is often very discontinuous and thus allows the passage of large particles; the second factor relies on the lack of lymphatic drainage in tumors that would normally remove these particles. Consequently, they can accumulate and remain in diseased tissues for a longer time than healthy ones.

For the EPR effect to occur, however, nanoparticles must remain long enough in the bloodstream to reach the tumor without being removed first by macrophages, as they are recognized as foreign agents by the immune system. It is therefore necessary to inhibit the phenomenon of opsonization, i.e., the recognition by the body the reticuloendothelial system of the foreign body; it occurs through the covering of this foreign body by specific components of the blood, such as opsonins, with subsequent elimination by macrophages. To avoid opsonization, many studies have shown that it is possible to coat nanosystems with polyethylene glycol (PEG) at a concentration of at least 0.1% [94,95,96].

In addition to the EPR effect, also known as passive target (the NP is administered with no modifications), the active target allows nanoparticles to reach and selectively bind cancer cells [94]. To obtain an active target, the most used technique is the binding on the NP surface of an agent (peptides, proteins, oligonucleotides, monoclonal antibodies) that interacts with specific receptors or that is recognized by the cancer cells [94,95,96]. Following this targeted drug delivery, higher drug concentrations can be released to the desired body location (organ, tissue, specific cells, intracellular organelles), avoiding side effects and reducing systemic toxicity [97].

Relating to the delivery modalities (ability or not to target specific cells), nanocarriers can be divided into three generations of compounds. “First generation” nanocarriers are able to accumulate by passive mechanisms through EPR effect (i.e., extravasation through gaps in tumor neovasculature). Among the first-generation vectors, liposomes-based drug delivery is the most successfully used in the clinic, as demonstrated by liposomal doxorubicin for breast, ovarian carcinoma and Kaposi’s sarcoma. The “second generation” of therapeutic nanovectors are characterized by additional functions; the surface modification with ligands allows their binding to specific molecules on tumor cells (active target mechanism) [98]. Moreover, they can be employed to co-deliver drugs and imaging agents, or they can be modified in order to have a controlled or triggered release. The so-called “third generation” nanovectors are based on a multi-stage strategy to improve targeting activity. These carriers are composed of different nanoparticles into a single vector to build a system that can cross biological barriers and induce selective tumor cytotoxicity [99]. An example is represented by biodegradable silicon nanoparticles [100,101]. Table 1 summarizes the characteristics of different nanomaterials, active or passive targeting and their advantages and disadvantages.

The above-mentioned strategies allowed the developing of more than 200 products that have been approved or are under clinical investigation [102]. In contrast with the large number of drug delivery systems that were shown to be successful at preclinical stages, recent studies demonstrate that clinical translation is a challenging process with about 10% of success in approval rate for therapeutics entering in phase I trials [103].

## 3. Natural Products and Drug Delivery in Oncology

Natural products represent a large family of different chemical entities with a wide variety of activities and pharmacological effects [104]. They originate from bacterial, fungal, plant and marine animal sources [105] and have several applications in different sectors such as food, agricultural [106], pharmaceutical [107], packaging [108] and cosmetics [109]. They are often used as flavorings, beverages, repellents and fragrances as well as for their medicinal purposes [110].

In recent years, biomedical research has focused its attention at searching substances of natural origin as possible chemosensitizing and chemopreventive agents [111,112,113]. Indeed, most of the anticancer drugs employed in therapy derive from natural substances or are related to them. In addition, the molecular diversity of these products with great biological potential have yet to be studied and discovered [114,115].

In 1940, the first antitumor antibiotic, actinomycin D, was isolated from the fungus *Actinomyces antibioticus* [116] Since then, many substances of natural origin have been subjected to in vitro and in vivo studies to assess their ability to improve the therapeutic index of chemotherapy. Taxanes, derived from plants belonging to the genus Taxus, have been extensively studied. Paclitaxel (PTX or Taxol^®^), collected from the bark of *Taxus brevifolia* [117], is a semi-synthetic form of taxane and acts as a microtubule-stabilizing drug inducing mitotic arrest and cell death. Nowadays, the drug represents a first-line treatment for ovarian, breast, lung and colon cancer and a second-line treatment for AIDS-related Kaposi’s sarcoma [118,119]. Dr. Bernardo’ research group conducted clinical trials to demonstrate the effectiveness of paclitaxel nanoparticles, linked to albumin, to treat advanced breast cancer [120].

There are many in vitro and in vivo studies focusing on drug delivery systems of natural compounds in the field of oncology aimed at improve their solubility, bioavailability and selectivity [23,121,122,123].

Liposomes represent one of the most employed nanoparticle systems for cancer therapy. They are able to encapsulate both lipophilic and hydrophilic compounds within their phospholipid bilayer and the inner core. Many studies have shown that encapsulating natural substances in liposomes can improve their biological activity compared to non-encapsulation [124]. One example is represented by doxorubicin: similar to other anthracycline compounds, the free drug induces severe cardiotoxicity in many patients [125], and its encapsulation is able to decrease free doxorubicin toxicity. A lot of in vitro and preclinical studies have employed doxorubicin encapsulated inside micelle, metallic nanoparticles and also nanodiamonds. These formulations also demonstrated a slower drug plasma clearance, enhanced circulation and half-life [126,127,128,129].

To date, many nanodelivery systems have been developed and reported to effectively bypass MDR both in vitro and in vivo. The encapsulation of chemotherapeutics can avoid their direct interaction with ABC transporters both at plasma and intracellular membranes, thus modifying the intracellular drug concentration and localization. In addition to bypassing drug transporter activity, nanoparticle delivery systems can also be effective in inducing apoptotic cell death [130].

Cheng et al. [131] reported a co-delivery system based on cationic amphiphilic polyester PHB-PDMAEMA to release an anticancer drug (paclitaxel) and a therapeutic plasmid (Bcl-2 convertor Nur77/ΔDBD gene) to overcome both ABC transporter-related and not-related drug-resistance in liver cancer cells. This system was able to partially inhibit P-glycoprotein activity (reducing PTX efflux) and activate apoptotic function linked to Bcl-2 activity.

Nano-based systems are increasingly employed to simultaneously deliver multiple natural compounds in order to overcome MDR-associated side effects. In a recent in vitro and in vivo study, resveratrol and doxorubicin were co-encapsulated in poly(lactic-co-glycolic acid) (PLGA)-based nanoparticles. Resveratrol is a natural stilbene and a non-flavonoid polyphenol, present in grapes, peanuts and red wine. This phytoestrogen possesses anti-oxidant, anti-inflammatory, cardioprotective and anti-cancer properties [132]. The PLGA-based nanoparticle system was able to simultaneously deliver both compounds into the nucleus of doxorubicin-resistant human breast cancer cells, thus increasing cell cytotoxicity. Nanoparticles employed overcame doxorubicin resistance by inhibiting the expression of P-gp, MRP-1 and BCRP drug transporters, and inducing apoptosis through NF-κB and Bcl-2 downregulation. In addition, this delivery system was also effective in inhibiting in vivo tumor growth with no significant induction of systemic toxicity [133]. In another recent study, docetaxel and resveratrol were encapsulated in planetary ball-milled nanoparticles bearing on the surface folic acid to be delivered in human cancer prostate cells, which highly express folate receptor on cell membrane. Cell treatments resulted in upregulation of pro-apoptotic (Bax, Bak) and downregulation of anti-apoptotic genes (Bcl-2, Bcl-Xl); furthermore, encapsulated drugs induced a significant downregulation of some ABC transporter genes (ABCB1, ABCC1 and ABCG2) at both mRNA and protein (P-glycoprotein, MRP1 and BCRP, respectively) levels in docetaxel-resistant cells, contributing to MDR phenotype reversal [134].

The plant alkaloid voacamine, isolated from the bark of the *Pescheria fuchsiaefolia* tree, has been demonstrated to enhance doxorubicin cytotoxicity and induce chemosensitizing effect on cultured multidrug-resistant U-2 OS-DX osteosarcoma and melanoma cell line Me30966 when used at noncytotoxic concentrations [135]. Voacamine encapsulated into different cationic liposome formulations was more efficient than free molecule to revert resistance of osteosarcoma cells resistant to doxorubicin [136].

In a study by Gupta et al., synthesized chitosan-based nanoparticles loaded with PTX were employed for the treatment of different kinds of cancer. Nanoparticle-loaded drug exhibited better activity with sustained release, higher cell uptake and reduced hemolytic toxicity in MDA-MB-231 breast cancer cell lines compared with pure PTX [23,137].

Maeng et al. demonstrated that folate-functionalized superparamagnetic iron oxide nanoparticles, developed previously for liver cancer cure, are also been used for the delivery of Doxil^®^. The in vivo studies showed a decrease of tumor volume compared with Doxil^®^ alone, while folate aided and enhanced specific targeting. These results indicate that this nano-drug is a promising candidate for treating liver cancer and monitoring the progress of the cancer using magnetic resonance imaging [138].

Curcumin is a natural polyphenolic compound extracted from the plant turmeric; a lot of in vitro and some in vivo studies demonstrated its anticancer properties in breast, prostate, bone, cervices, lung and liver cancer cell lines. Unfortunately, free curcumin is not soluble in water and not very bioavailable, features that limit its application in the clinics; the encapsulation in nanoparticles such as liposomes has solved these problems [139,140,141].

Cheng et al. have developed a method based on the self-assembly behavior of phospholipid in water and the diffusion of curcumin into liposomal membranes driven by hydrophobic forces. This method has allowed a good encapsulation and bio-accessibility of curcumin for its applications in cancer treatment [142].

Liposomal curcumin (Lipocurc^TM^) can reduce the tumor growth of pancreatic and colorectal cancer. In a phase I dose escalation study the safety, pharmacokinetics, tolerability and activity of intravenously administered liposomal curcumin were evaluated in patients with locally advanced or metastatic cancer. The results showed that 300 mg/m^2^ liposomal curcumin over 6 h was the maximum tolerated dose in these heavily pretreated patients, and is the recommended starting dose for anti-cancer trials [143,144].

Another study of Madamsetty et al. showed the remarkable anti-tumor efficacy of PEGylated nanodiamonds carrying a dual payload of irinotecan (a derivate of camptothecin that inhibits the action of topoisomerase I) plus curcumin. These results highlight the potential use of such nano-carriers in the treatment of patients with pancreatic cancer [145].

Artemisinin was discovered by Youyou Tu in 1972 and is used as an antimalarial for the treatment of multi-drug resistant strains of *Plasmodium falciparum* that causes malaria infection. Artemisinin is a sesquiterpene lactone obtained from sweet wormwood, Artemisia annua [146]. This substance and its derivatives (artemisinins) show good antitumoral activity. Artemisinin is loaded in many nanocarriers such as liposomes, niosomes, micelles, solid lipid nanocarriers, nanostructured lipid carriers, nanoparticles, fullerenes and nanotubes with different therapeutic applications. Nowadays, there are many studies on Artemisinin and its encapsulation inside nanoparticles to improve drug delivery and to increase blood circulation, as the therapeutic value of Artemisinin is limited due to a low bioavailability and a short half-life [147,148].

Leto et al. investigated the antitumor properties of Artemisinin encapsulated in a PEGylated nanoliposome decorated with transferrin receptors on HCT-8 cell line. The results confirmed the enhanced delivery of Artemisinin loaded in liposomes actively targeted with transferrin in comparison with the other Artemisinin-loaded liposomes and an improved cytotoxicity [149].

Many in vitro studies on resveratrol have reported its involvement in different cellular responses, such as cell cycle arrest, induction of differentiation, apoptosis and growth inhibition in several types of cancer, principally prostate and colon cancers [133]. Unfortunately, resveratrol displays low bioavailability, low water solubility and instability. Consequently, a lot of studies based on nanoformulations are trying to avoid these negative features. It was loaded inside mPEG poly (epsiloncaprolactone) based nanoparticles [150] to increase cell death of glioma cells in vitro [151]. Caddeo et al. studied the effect of resveratrol incorporated in liposomes on proliferation and UVB protection of cells. The results demonstrated that liposomes prevented the cytotoxicity of resveratrol at high concentrations, avoiding its immediate and massive intracellular distribution, and increased the ability of resveratrol to stimulate the proliferation of the cells and their ability to survive under stress conditions caused by UV-B light [152].

Other natural substances with promising anticancer properties are essential oils (EOs). They are a complex mixture of hydrophobic and volatile compounds synthesized from aromatic plants. They are constituted of terpenoids, phenol-derived aromatic components and aliphatic components. EOs have been widely used for their antimicrobial, antioxidant, anti-inflammatory, immunomodulatory and anticancer properties in vitro [153]. The characterization of EOs is made difficult by their complexity and by the different compositions present in the same oil having different geographical origins. Moreover, they are volatile; thus, they can easily decompose, owing to direct exposure to heat, humidity, light and oxygen. Encapsulation of EOs in micro or nanometric systems is an interesting strategy to provide better stability to the volatile compounds and protect them against environmental factors that may cause chemical degradation. In addition, it can increase EOs bioactivity, decreasing their volatility [154].

An in vivo study on mice investigated the oral delivery of frankincense and myrrh oil (FMO) loaded in Solid lipid nanoparticles (SLNs), a new nanoparticle-based drug-delivery system with particles that range in diameter from 10 to 1000 nm (FMO-SLNs). Frankincense and myrrh are gum resins obtained from the genera *Boswellia* and *Commiphora*. Frankincense and myrrh oil exhibit many biological activities such as antimicrobial, anti-inflammatory and antitumor. The study of Shi et al. focused on the preparation of SLNs, which are capable of improving the stability and antitumor efficacy of FMO. The results demonstrated that SLNs can be used to increase the stability and/or to improve the in vivo antitumor efficacy of FMO [155].

Carvacrol is a monoterpene present in the essential oil of oregano. It has been studied for its therapeutic properties, especially in the control of painful conditions and inflammation during cancer condition. The encapsulation of carvacrol in a complex of β-cyclodextrin and administered (50 mg/kg, orally) in mice with sarcoma tumor reduced hyperalgesia. However, pure carvacrol did not cause significant changes in nociceptive responses. These results produced evidence that the encapsulation of carvacrol in β-cyclodextrin can be useful for the development of new options for pain management [156].

*Aloe emodin* (1,8-dihydro-3-hydroxymethyl-anthraquinone, AE) is one of the constituents of *Aloe vera* gel. It shows multiple properties (antifungal, antibacterial, antiviral activities, liver protective) [157]. A lot of in vitro studies have demonstrated the ability of AE to reduce the viability and proliferation of different cancer cell lines, induce the apoptotic cell death, inhibit adhesion and migration process [158,159,160,161,162] and also promote cell differentiation [163]. Unfortunately, AE application in anticancer therapies might be hindered by its scarce solubility in aqueous environment. Several studies are currently searching for AE loading into nanocarriers, to obtain a selective delivery to target sites. Cationic liposomes (gemini-based) showed to efficiently load AE (which possesses a weak acid nature) in their internal aqueous phase, in response to a pH difference between the inside and outside of the liposomes [164].

Li et al. investigated the effect of transfection of r-caspase-3 with nanoliposomes loaded with AE and photodynamic therapy in human gastric cancer cells [165]. Nano-AE was employed as delivery carrier for plasmid r-caspase-3 transfection, which, consequently, induced apoptosis, and also as photosensitizer for photodynamic therapy. This nanocarrier system was demonstrated to increase the AE solubility and improve its bioavailability, without altering its properties. Some of the natural products and delivery systems described above are listed in Table 2.

## 4. FDA Approved Nanodevices

Currently, many nanomedicines that are used for cancer therapy and approved by the U.S. Food and Drug Administration (FDA), are extracted from natural compounds, such as Doxorubicin, Paclitaxel, Vincristin, Kaempferol, Silamarin, Resveratrol and Curcumin [23,168,169,170,171].

As known, the first nano-drug FDA-approved was a PEGylated liposomal formulation of doxorubicin, Doxil^®^/Caelyx^®^ (1995) [172], which successfully improved outcomes with an enhancement of drug concentrations in the tumor, when compared with free drug, with a concomitant reduction in cardiotoxicity [122].

Myocet^TM^ is a non-pegylated liposomal formulation loaded with doxorubicin. It is used to treat women with metastatic breast cancer usually in combination with cyclophosphamide. It has received a marketing authorization valid throughout the EU on 13 July 2000. Myocet^TM^ shows reduced cardiotoxicity and more prolonged intratumor accumulation, as compared to free drug, thanking to the citrate complex and the loading method through a pH gradient [173].

DaunoXome^®^ is another natural nanodrug currently used in clinical mostly for advanced HIV-associated Kaposi’s sarcoma. It is an aqueous solution of the citrate salt of daunorubicin encapsulated into liposomes. Daunorubicin is an anthracycline antibiotic with antineoplastic activity, obtained from *Streptomyces peucetius* and *Streptomyces coeruleorubidus*. Free daunorubicin was employed for acute lymphocytic leukemia and acute myeloid leukemia and induces many adverse effects such as cardiovascular, dermatologic (alopecia), gastrointestinal and genitourinary. If administered as DaunoXome^®^, daunorubicin displays substantially altered pharmacokinetics with reduced side effects [174].

ThermoDox^®^ uses lysolipid thermally sensitive liposome to encapsulate doxorubicin. Heat-sensitive liposomes rapidly change structure when heated to 40–45 °C, creating openings in the liposome, thus allowing doxorubicin delivery directly into and around the targeted tumor. ThermoDox^®^ is used for primary liver cancer and it is being evaluated in a two-arm, double-blinded, placebo-controlled, randomized phase 3 clinical study, designed to assess ThermoDox^®^ in combination with standardized radiofrequency thermal ablation (RFA ≥ 45). The goal of this study is to increase the effective treatment zone in order to capture micrometastases [175].

Another nano-drug FDA-approved is the vincristine sulfate liposome called Marqibo^®^, employed for treatment of hematologic malignancies and solid tumors. This nano-drug is a sphingomyelin and cholesterol-based nanoparticle formulation of Vincristine that is able to overcome the dosing and pharmacokinetic limitations, increase the circulation time, optimize delivery to target tissues and facilitate dose intensification without increasing toxicity compared to drug free [176].

A clinical/phase III study of another nano-drug is based on nanoparticle drug delivery formulation that encapsulates PTX in polymeric micelles. It is an open-label phase III non-inferiority trial to compare the efficacy and safety of the nano-drug called NK105 and PTX in metastatic or recurrent breast cancer. The primary endpoint was not met, but NK105 had a better peripheral sensory neuropathy toxicity profile than free PTX [166].

An Australian clinical/Phase I open-label single-ascending study is still ongoing to measure the safety, tolerability and pharmacokinetics of OP-101 (Dendrimer N-acetyl-cysteine), normally used for the treatment of cell or squamous cell skin cancer, after subcutaneous administration in healthy volunteers. The trial includes two outcomes: primary outcome evaluates the safety and tolerability of OP-101 after single subcutaneous doses in healthy subjects (adverse events), which include laboratory test variables; the secondary outcome is to determine the pharmacokinetic profile of OP-101 after single doses in healthy subjects (plasma and urine concentrations). Recently, the US FDA approved to initiate phase II trial in COVID-19 infections [167].

Abraxane^®^ is the clinically approved albumin nanoparticle formulation of PTX, in the form of nanospheres (130 nm in diameter). This formulation has increased the bioavailability of paclitaxel, resulting in higher intratumor concentrations facilitated by albumin receptor (gp60)-mediated endothelial transcytosis. This represents a very successful drug delivery system of PTX, approved in different countries for the treatment of metastatic breast cancer, advanced pancreatic cancer and for advanced non-small cell lung cancer in combination with carboplatin for patients who are not candidates for curative surgery or radiation therapy [177].

The FDA approved nanodevices described in this section and the corresponding reference are summarized in Table 3.

## 5. Conclusions

Traditional approaches for cancer treatment are sometimes limited because of their low specificity, responsible for severe side effects and toxicity, and the possible induction of the MDR phenotype. As a consequence, the searching for more effective anticancer therapeutic strategies is increasingly driven by the need to selectively kill cancer cells, defeat the MDR phenomenon and increase the specificity of a drug through the spatial, temporal and dose control of its release.

Natural products represent an important source for the discovery of novel anticancer drugs to be used both at a preventive and therapeutic level. Unfortunately, some of the natural compounds’ features restrict their application in anticancer therapy. A lot of studies have already demonstrated that strategies based on nanomedicine and nanodelivery systems allow to improve the efficacy of natural compounds with anticancer properties, increasing their solubility, bioavailability and selectivity, reducing their systemic toxicity and in some cases circumventing a drug resistance response. Nowadays, there are different types of drug carriers with unique and versatile properties that make them appropriate for oncology applications. Increasingly advanced research allows to design multifunctional nanocarriers, characterized by a gradual and selective release of multiple compounds at specific tumor sites. NPs carrying and delivering both compounds already used in chemotherapy and natural products are demonstrated in in vitro and preclinical studies to be very effective for both therapeutic and chemosensitizing purposes.

## Figures and Tables

**Figure 1 molecules-25-04560-f001:**
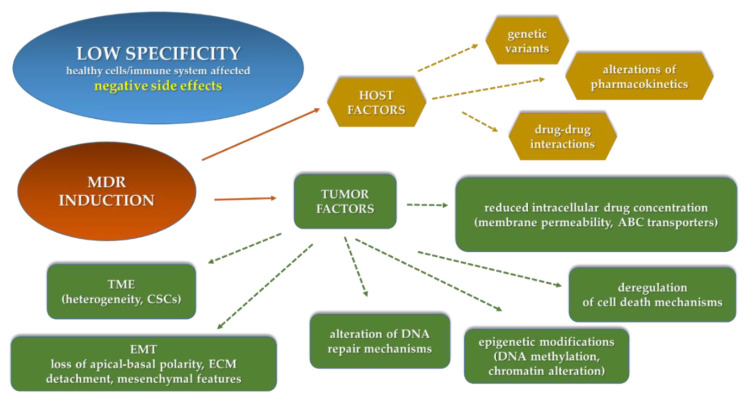
Schematic representation of the main mechanisms responsible for anticancer therapy failure (ABC, ATP-Binding Cassette; MDR, multidrug resistance; TME, tumor microenvironment; CSCs, cancer stem cells; EMT, epithelial-mesenchymal transition; ECM, extracellular matrix).

**Figure 2 molecules-25-04560-f002:**
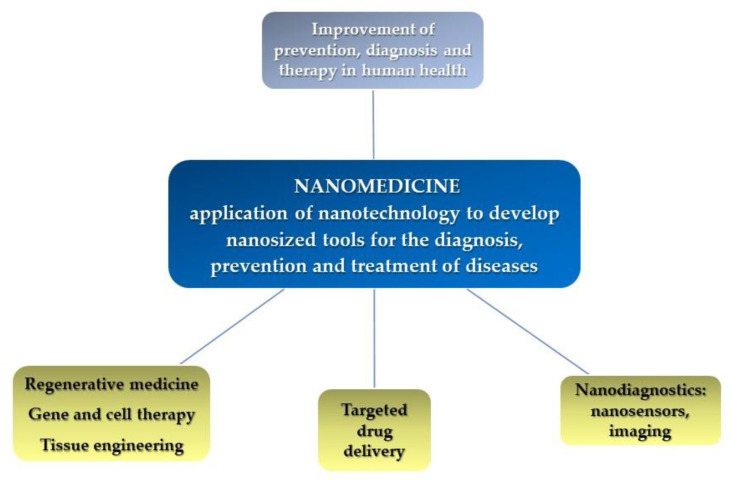
Nanomedicine definition.

**Figure 3 molecules-25-04560-f003:**
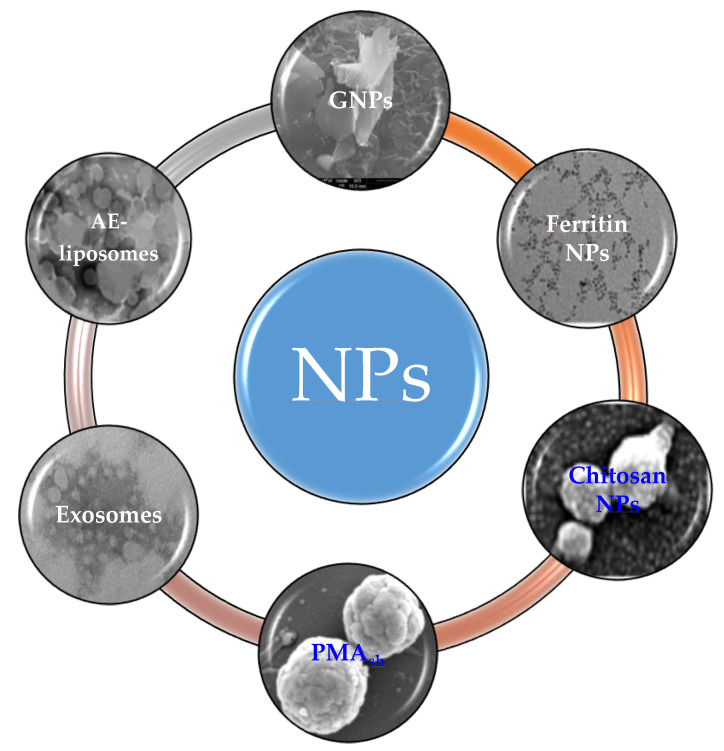
Schematic representation of nanoparticle (NP) categories. AE, Aloe emodin; GNPs, Graphite NanoPlatelets; PMA_sh_, Redox-active microcapsules based on thiolated poly (methacrylic acid). Micrographs are from collection of National Center for Drug Research and Evaluation, Istituto Superiore di Sanità, Rome, Italy.

**Table 1 molecules-25-04560-t001:** Main characteristics of nanomaterials.

Nanomaterials	Passive Targeting	Active Targeting	Advantages	Disadvantages
Lipid based NPs	Accumulate through the EPR effect	Possibility to decorate with specific ligand	Biocompatible, biodegradable, reduced toxicity	High Clearance via RES
Carbon based NPs	Promoting increased accumulation in tumor sites	Lower toxicity, increased efficacy	Biocompatible	Immunogenic, thrombotic
Polymeric NPs	Prolonged circulation times	Higher drug concentration in tumor sites	Easy design, wide shape variabilities	Induction of inflammatory processes
Metallic and Magnetic NPs	Combination of diagnosis and treatment	Involved in multimodal cancer treatment to enhance drug accumulation	No specific drug distribution	Possible high toxicity and low stability and biocompatibility

RES, reticuloendothelial system. EPR: enhanced permeability and retention.

**Table 2 molecules-25-04560-t002:** Natural products and their delivery systems.

Natural Product	NPs	Evaluation	Ref.
Paclitaxel (PTX)	Cationic nanoparticle complex	in vitro	[131]
Resveratrol plus Doxorubicin	Poly(lactic-co-glycolic acid) (PLGA)-based nanoparticles	in vitro	[133]
Resveratrol plus Docetaxel	Planetary ball milled (PBM) nanoparticles	in vitro	[134]
PTX	Chitosan-based nanoparticles	in vitro	[137]
Curcumin	Liposomes	in vitro	[142]
Artemisinin	PEGylated nanoliposomes	in vitro	[149]
Resveratrol	mPEG poly (epsiloncaprolactone) nanoparticles	in vitro	[151]
Resveratrol	Liposomes	in vitro	[152]
Doxil^®^ (Doxorubicin)	Iron oxide nanoparticles	in vivo	[138]
Irinotecan plus Curcumin	PEGylated nanodiamonds	in vivo	[128]
Frankincense and Myrrh oil (FMO)	Solid lipid nanoparticles	in vivo	[155]
Carvacrol	Complex of β-cyclodextrin	in vivo	[156]
PTX	Polymeric micelles NK105	Clinical trial	[166]
PTX plus mAb anti-HER2+	Pegylated Immunoliposomes	Clinical trial	[32]
Doxo plus mAb 2C5	Liposomes	Clinical trial	[33]
N-acetyl-cysteine	Dendrimer OP-101	Clinical trial	[167]

**Table 3 molecules-25-04560-t003:** A summary of FDA approved nanodevices.

FDA Approved	Ref.
Doxil^®^/Caelix^®^	[172]
Marqibo^®^	[176]
Myocet^®^	[173]
Abraxane^®^	[177]
DaunoXome^®^	[174]
ThermoDox^®^	[175]
Lipocurc^TM^	[143,144]

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
