# Peer review of "Drug Delivery Systems of Natural Products in Oncology"

_molecules, 2020, doi:10.3390/molecules25194560_

Round 1

Reviewer 1 Report

The review focuses on the use of natural products with anti-tumoral effects encapsulated into different delivery systems in order to enhance their anticancer effects, increase stability and bioavailability while reducing side effects and improving specificity.

It is a really nice and very well written overview. However, some schemes, figures/data taken from the cited papers as well as three resuming tables have to be inserted to make the manuscript more elaborate and to improve the readability which certainly will attract a broad readership. First, a resuming table in relation to the characteristics of the different nanomaterials, active or passive targeting and advantages and disadvantages of them. Second, a resuming table concerning the many in vitro and in vivo studies focusing on drug delivery of natural compounds and the type of nanomaterials used. And third, a table about the FDA approved nanodevices.

Author Response

Dear Reviewer,

We have carefully revised our manuscript and addressed all your comments. The replies to  comments are given below.

Response to Reviewer 1 Comments

Comment-1: First, a resuming table in relation to the characteristics of the different nanomaterials, active or passive targeting and advantages and disadvantages of them. Second, a resuming table concerning the many in vitro and in vivo studies focusing on drug delivery of natural compounds and the type of nanomaterials used. And third, a table about the FDA approved nanodevices.

Response 1: Thank you for your valuable suggestion. We have added three new tables in the manuscript: Table 1 (characteristics of the different nanomaterials, active or passive targeting and advantages and disadvantages ) Page 11 (Line 365);

Table 2 ( the many in vitro and in vivo studies focusing on drug delivery of natural compounds and the type of nanomaterials used) Page 16 (Line 553);

Table 3 (FDA approved nanodevices) Page 18 (Line 615).

Best Regards

Reviewer 2 Report

In this review manuscript, the author discusses a recent development in the delivery of natural compounds for cancer treatment. Nevertheless, similar reviews have been recently published, this review brings new insight into the area. The manuscript is well organized and written, therefore, I recommended its publication. 

Here are more comments from my site:

The authors should discuss the difference between the manuscripts that are already existing in the literature:

  1. Nano based drug delivery systems: recent developments and future prospects (J Nanobiotechnol (2018) 16:71.)
  2. Natural product-based nanomedicine: recent advances and issues (Int J Nanomedicine. 2015; 10: 6055–6074.)

Page 5, line 146: The polymeric nanoparticles should be added to the scheme.

Page 8, line 279: The supramolecular polylactides were also used to overcome MDR of cancer cells (European Polymer Journal, 2018, 109, 117-123). It should be discussed in the paper.

Page 9. Line 30: Fe304???

Line 313: Thanking to????? the enhanced

Page 13, line 474: [The results ???

Page 14, line 499: b-cyclodextrin ???

Author Response

Dear Reviewer,

we have carefully revised our manuscript and addressed your comments.  The replies to  comments are given below.

Comment-1:The authors should discuss the difference between the manuscripts that are already existing in the literature:

Nano based drug delivery systems: recent developments and future prospects (J Nanobiotechnol (2018) 16:71.)

Natural product-based nanomedicine: recent advances and issues (Int J Nanomedicine. 2015; 10: 6055–6074.)

Response 1: Thank you for your valuable suggestion, the authors have discussed the difference between the manuscripts that are already existing in the literature in Page 4 (Line 119-124).

Comment-2: Page 5, line 146: The polymeric nanoparticles should be added to the scheme.

Response 2: The authors have added to the scheme PMAsh nanoparticles.

Comment-3: Page 8, line 279: The supramolecular polylactides were also used to overcome MDR of cancer cells (European Polymer Journal, 2018, 109, 117-123). It should be discussed in the paper.

Response 3: We have discussed and added this reference  (n. 76) in the text Page. 8 (Line 279-283).

Comment-4: Page 9. Line 30: Fe304???

Response 4: Page 9 Line 311: Fe3O4  Correct

Comment-5: Line 313: Thanking to????? the enhanced

Response 5: Page 9 Line 318: Thanks to  Correct

Comment-6: Page 13, line 474: [The results ???

Response 6: Page 14, line 491: Deleted bracket

Comment-7: Page 14, line 499: b-cyclodextrin ???.....

Response 7: Page 15, line 516 β-cyclodextrin  Correct

We have tried to respond to all requests from the referees.

We hope that the edited manuscript conforms to the standard required for publication in Molecules.

Thank you for your attention to this topic.

Best regards